# A *Streptomyces* sp. NEAU-HV9: Isolation, Identification, and Potential as a Biocontrol Agent against *Ralstonia solanacearum* of Tomato Plants

**DOI:** 10.3390/microorganisms8030351

**Published:** 2020-03-01

**Authors:** Ling Ling, Xiaoyang Han, Xiao Li, Xue Zhang, Han Wang, Lida Zhang, Peng Cao, Yutong Wu, Xiangjing Wang, Junwei Zhao, Wensheng Xiang

**Affiliations:** 1Key Laboratory of Agricultural Microbiology of Heilongjiang Province, Northeast Agricultural University, No. 59 Mucai Street, Xiangfang District, Harbin 150030, China; LLYNL2621161093@163.com (L.L.); hanxy139251@163.com (X.H.); Lx1244070003@126.com (X.L.); zhangxue_968425@163.com (X.Z.); wanghan507555536@gmail.com (H.W.); yone910310@163.com (L.Z.); cp511@126.com (P.C.); 18103699151@163.com (Y.W.); wangneau2013@163.com (X.W.); 2State Key Laboratory for Biology of Plant Diseases and Insect Pests, Institute of Plant Protection, Chinese Academy of Agricultural Sciences, Beijing 100193, China

**Keywords:** antibacterial activity, *Ralstonia solanacearum*, *Streptomyces* sp. NEAU-HV9, actinomycin D

## Abstract

*Ralstonia solanacearum* is an important soil-borne bacterial plant pathogen. In this study, an actinomycete strain named NEAU-HV9 that showed strong antibacterial activity against *Ralstonia solanacearum* was isolated from soil using an in vitro screening technique. Based on physiological and morphological characteristics and 98.90% of 16S rRNA gene sequence similarity with *Streptomyces panaciradicis* 1MR-8^T^, the strain was identified as a member of the genus *Streptomyces*. Tomato seedling and pot culture experiments showed that after pre-inoculation with the strain NEAU-HV9, the disease occurrence of tomato seedlings was effectively prevented for *R.*
*solanacearum*. Then, a bioactivity-guided approach was employed to isolate and determine the chemical identity of bioactive constituents with antibacterial activity from strain NEAU-HV9. The structure of the antibacterial metabolite was determined as actinomycin D on the basis of extensive spectroscopic analysis. To our knowledge, this is the first report that actinomycin D has strong antibacterial activity against *R. solanacearum* with a MIC (minimum inhibitory concentration) of 0.6 mg L^−1^ (0.48 μmol L^−1^). The in vivo antibacterial activity experiment showed that actinomycin D possessed significant preventive efficacy against *R. solanacearum* in tomato seedlings. Thus, strain NEAU-HV9 could be used as BCA (biological control agent) against *R. solanacearum*, and actinomycin D might be a promising candidate for a new antibacterial agent against *R. solanacearum*.

## 1. Introduction

Tomato is one of the world’s most important vegetable crops, with a global annual yield of approximately 160 million tons [1,2]. In China, long term continuous cropping is the main planting practice for tomato, which has led to serious soilborne diseases [3]. *Ralstonia solanacearum* [4] is an important soilborne bacterial plant pathogen [5]. Bacterial wilt caused by *R. solanacearum* is a serious and common disease, which reduces the yield of tomato and many other crops in tropical, subtropical, and warm-temperature regions of the world [6]. Because of worldwide distribution and a large host range of more than 200 plant species in 50 families, including pepper, tomato, tobacco, potato, peanut, and banana, this soil bacterium has been recognized as one of the causative agents of bacterial wilt disease and is one of the leading models in pathogenicity [5]. In the absence of host plants, this bacterium can be free-living as a saprophyte in the soil or in water [7]. Plant breeding, field sanitation, crop rotation, and use of bactericides have met with only limited success for *R. solanacearum* [8]. Furthermore, pathogenic microbial multi-drug resistance is also increasing. Therefore, new natural resources and antibiotics for suppressing this soilborne disease are needed.

Various recent studies have showed that biological control of bacterial wilt disease could be achieved using antagonistic bacteria [8,9]. The suppressive effect of some antagonistic bacteria on *R. solanacearum* was reported by Toyota and Kimura [10]. Moreover, the use of antagonistic bacteria to be effective in control of *R. solanacearum* has been proved by Ciampi-Panno et al. under field conditions [8]. *Streptomycetes* are gaining interest in agriculture as plant growth promoting (PGP) bacteria and/or biological control agents (BCAs) [11,12]. The *Streptomyces* genus comprises Gram-positive bacteria which show a filamentous form; they can grow in various environments. Several *Streptomyces* species such as *S. aureofaciens*, *S. avermitilis*, *S. lividans*, *S. humidus*, *S. hygroscopicus*, *S. lydicus*, *S. plicatus*, *S. olivaceoviridis*, *S. roseoflavus*, *S. scabies* and *S. violaceusniger* have been used to control soilborne diseases due to their greatly antagonistic activities by production of various antimicrobial substances [13,14,15].

*Actinobacteria* are famous for producing a variety of natural bioactive metabolites. *Streptomyces* is an important source of bioactive compounds among all members of antibiotic production, accounting for two-thirds of commercially available antibiotics [16]. Actinomycins belonging to a family of chromopeptide lactones are produced by various *Streptomyces*. Among several antibiotics produced by this genus, actinomycins are prominent. More than 20 naturally-occurring actinomycins were isolated and observed to have commonality of two pentapeptidolactone moieties with an actnoyl chromophore [17]; however, they differ in functional and/or positional group. Among actinomycins, actinomycin D has been widely studied and used clinically as an anticancer drug, especially in the treatment of childhood rhabdomyosarcoma, infantile kidney tumors and several other malignant tumors [18,19]. However, no reports have been published on actinomycin D against phytopathogen *R. solanacearum*.

In the existing protocol for virulence assays, one-month old tomato plantlets are soil-inoculated with the bacterium and wilting symptoms, if any, are observed and recorded. In usual ground work, tomato seeds are sown to obtain seedlings that take 5–6 days to sprout. Seedlings are then transferred to pots containing soil and grown in a greenhouse for about one month. Following this, plants are shifted to a growth chamber where plants are inoculated with the pathogen by soil drench or the stem inoculation method [20,21]. Using this approach, it usually takes 40 days to perform a single virulence assay. The infection achieved in this way is generally not axenic as the soil conditions used are not devoid of other bacterial communities that can colonize the plant during its growth prior to the infection study. Singh et al. [22] described a simple assay to study the pathogenicity of *R. solanacearum* on freshly grown tomato seedlings instead of fully-grown tomato plants. From seed germination to completion of the infection process, the study takes around 15 to 20 days. Pathogenicity due to *R. solanacearum* was also demonstrated when there is no significant plant growth since no mineral/growth inducing factors have been added into the water [23]. Under this same condition, there are reports of the bacterium’s survivability without any growth [24]. The death of tomato seedlings was actually occurring due to the presence of *R. solanacearum* in the water. On the basis of the previous study, we have discussed an approach to study biological assays in tomato seedlings.

In this study, a *Streptomyces* sp., NEAU-HV9, was isolated and showed strong antimicrobial activity against *R. solanacearum*. The taxonomic identity of NEAU-HV9 was determined by a combination of 16S rRNA gene sequence analysis with morphological and physiological characteristics. The potential control of actinomycin D produced by the strain NEAU-HV9 against *R. solanacearum* was also investigated.

## 2. Materials and Methods

### 2.1. Sample Collection

Soil samples were collected from a field situated in Bama yao Autonomous County, Hechi City, Guangxi zhuang Autonomous Region (24°15′ N, 107°26′ E). The collected soil samples were brought to the laboratory in sterile bags and kept at 4 °C until further analysis. Before isolation of actinomycetes, the soil samples were air-dried at room temperature.

### 2.2. Screening and Isolation of Actinomycetes

The soil sample (5 g) was mixed with 45 mL distilled water and followed by an ultrasonic treatment (160 W) for 3 min. The soil suspension was incubated at 28 °C and 250 rpm on a rotary shaker for 30 min. Subsequently, the supernatant was collected and subjected to serial dilutions from 10^−2^ to 10^−5^. Each dilution (200 µL) was spread on a plate of humic acid-vitamin (HV) agar [25] supplemented with cycloheximide (50 mg L^−1^) and nalidixic acid (20 mg L^−1^). Colonies were transferred and purified on International *Streptomyces* Project (ISP) medium 3 [26] and stored for a long time in glycerol suspensions (20%, *v*/*v*) at −80 °C after 14 days of aerobic incubation at 28 °C.

### 2.3. Screening of Antagonistic Actinobacteria Strains

The isolates were screened using the agar well diffusion method, and *R. solanacearum* was used as the indicator bacterium [27]. To further investigate the antibacterial components produced by the isolated cultures, these strains were cultured in ISP 2 medium [26] and the inhibitory activities of the supernatant and cell precipitate were tested. Initially, the isolated cultures were grown in ISP 2 medium and incubated at 28 °C on a rotary shaker. After 7 days of incubation, the supernatants were obtained by centrifugation at 8000 rpm and 4 °C for 10 min and subsequently filtrated with a 0.2 µm membrane filter. The cell precipitates were extracted with an equal volume of methanol for approximately 24 h [28]. A cell suspension (1 mL at 1 × 10^8^ cfu mL^−1^) of *R. solanacearum* was aseptically plated onto Bactoagar-glucose (BG) media supplemented with 0.5% glucose [22]. Supernatant and methanol extracts were collected from each isolate and tested initially for antimicrobial activity against *R. solanacearum*; each well contained 200 μL of supernatant or methanol extract. The plates were incubated at 37 °C for 12 h to test antibacterial activity. The diameters of inhibition zones were measured by using vernier calipers [29]. The experiments were conducted twice. The isolates that showed activities against tested organisms were collected and maintained. Among the collected isolates, the potential isolate designated as NEAU-HV9 was selected for further studies.

### 2.4. Morphological and Biochemical Characteristics of NEAU-HV9

Morphological characteristics, using cultures grown on ISP 3 medium at 28 °C for 2 weeks, were observed by light microscopy (Nikon ECLIPSE E200, Nikon Corporation, Tokyo, Japan) and scanning electron microscopy (Hitachi SU8010, Hitachi Co., Tokyo, Japan). Scanning electron microscopy samples were prepared as described by Jin et al. [30]. Cultural characteristics were determined using 2-week cultures grown at 28 °C on Czapek’s agar [31], Bennett’s agar [32], Nutrient agar [33], ISP 1 agar and ISP 2-7 media [26]. The color designation of substrate mycelium and aerial mycelium was done with ISCC–NBS (Inter-Society Color Council-National Bureau of Standards) Color Charts Standard Sample No. 2106 [34]. Growth at different temperatures (10 °C, 15 °C, 18 °C, 20 °C, 25 °C, 28 °C, 32 °C, 35 °C, 37 °C and 40 °C) was determined on ISP 3 medium after incubation for 14 days. Growth tests for pH range (pH 4.0–10.0, at intervals of 1.0 pH unit) using the buffer system described by Zhao et al. [35] and NaCl tolerance (0%, 1%, 2%, 3%, 4%, 5%, 6%, 7%, 8%, 9% and 10%, w/v) were tested in ISP 2 broth at 28 °C for 14 days on a rotary shaker. Biochemical testing (decomposition of adenine, casein, hypoxanthine, tyrosine, xanthine and cellulose, hydrolysis of starch, aesculin and gelatin, milk peptonization and coagulation, nitrate reduction and H_2_S production), the utilization of sole carbon and nitrogen sources were examined as described previously [36,37].

### 2.5. Phylogenetic Analysis of NEAU-HV9

Strain NEAU-HV9 was cultured in ISP 2 medium for 3 days at 28 °C to harvest cells. The genomic DNA was isolated using a Bacteria DNA Kit (TIANGEN Biotech, Co. Ltd., Beijing, China). The universal bacterial primers 27F and 1541R were used to carry out PCR amplification of the 16S rRNA gene sequence [38,39]. The purified PCR product cloned into the vector pMD19-T (Takara) and sequenced by using an Applied Biosystems DNA sequencer (model 3730XL, Applied Biosystems Inc., Foster City, California, USA). The almost complete 16S rRNA gene sequence (1510 bp) was uploaded to the EzBioCloud server (Available online: https://www.ezbiocloud.net/) [40] to calculate pairwise 16S rRNA gene sequence similarity between strain NEAU-HV9 and related similar species. The phylogenetic tree was reconstructed with neighbor-joining trees [41] using MEGA 7.0 software [42]. The confidence value of branches of the neighbor-joining tree was assessed using bootstrap resampling with 1000 replication [43]. A distance matrix was calculated using Kimura’s two-parameter model [44]. All positions containing gaps and missing data were eliminated from the dataset (complete deletion option).

### 2.6. Fermentation

Strain NEAU-HV9 was grown and maintained for 7 days at 28 °C on ISP 3 medium agar plates. Fermentation involved the generation of a seed culture. The stock culture was transferred into two 250 mL Erlenmeyer flasks containing 50 mL of the ISP2 medium and incubated at 28 °C for 72 h on a rotary shaker at 250 rpm. All of the media were sterilized at 121 °C for 20 min. The seed culture (5%) was transferred into 75 flasks (250 mL) containing 100 mL of production medium. The production medium was composed of maltodextrin 4%, lactose 4%, yeast extract 0.5%, Mops 2% at pH 7.2–7.4. The flasks were incubated at 28 °C for 7 days, shaken at 250 rpm. The final 7.5 L fermentation broth was filtered to separate the supernatant and the mycelial cake. The supernatant was extracted with ethyl acetate three times (3 × 2 L), and the mycelial cake was extracted with MeOH (3 L). The organic phase was evaporated under reduced pressure at 55 °C to yield the red crude extract (5.2 g).

### 2.7. Isolation and Purification of Antibacterial Compounds

Crude extract from the mycelium and supernatant was combined and subjected to silica gel column chromatography (Qingdao Haiyang Chemical Group, Qingdao, China; 100–200 mesh; 100 × 3 cm column) using a gradient of ethyl acetate−MeOH (100:0−90:10) to yield three fractions (Fr.1-Fr.3) based on the TLC (thin layer chromatography) profiles. TLC was performed on silica-gel plates with solvent of ethyl acetate/MeoH (4:1). All fractions (Fr.) were screened against *R. solanacearum*. The most active, Fr.1 and Fr.2, were applied to a Sephadex LH-20 column eluted with CH_2_Cl_2_/MeOH (1:1, *v*/*v*) and then further purified by semipreparative HPLC (Agilent 1260, Zorbax SB-C18, 5 μm, 250 × 9.4 mm inner diameter; 1.5 mL/min; 220 nm; 254 nm; Agilent, Palo Alto, CA, USA) MeOH/H2O (90:10, *v*/*v*) to obtain Compound 1 (*tR* 10.928 min, 9.3 mg) and Compound 2 (*tR* 12.367 min, 60.4 mg). We chose the main product, Compound 2, for further research. NMR spectra (1H and 13C) were measured with a Bruker DRX-400 (400 MHz for ^1^H and 100 MHz for ^13^C) spectrometer (Bruker, Rheinstetten, Germany). The ESI-MS (electrospray ionization mass spectra) spectra were taken on a Q-TOF Micro LC-MS-MS mass spectrometer (Waters Co, Milford, MA, USA).

### 2.8. Determination of Minimum Inhibitory Concentration (MIC)

The minimum inhibitory concentration (MIC) of the antibacterial compounds was determined as described by Rathod et al. [45]. *R. solanacearum* was grown in BG medium with 0.5% glucose in shake flasks at 28 °C for 24 h. Cells were harvested by centrifugation, washed with 0.85% saline twice, then the supernatant was discarded and 0.85% saline was added to the washed cells. The suspensions were standardized to an optical density (OD) of 0.2 at 540 nm. Antibacterial compounds were two-fold serially diluted to obtain concentrations ranging from 0.2 to 12.8 mg L^−1^ and one tube without drug served as a control. All of the tubes were inoculated with 1 mL of suspension of *R. solanacearum* above and incubated at 37 °C for 12 to 16 h. The turbidity of each tube with respect to the control tube was measured. The MIC value was defined as the lowest concentration of a compound that completely inhibits growth.

### 2.9. Biological Assays in Tomato Seedlings

Germination of tomato seedlings and preparation of bacterial inoculum were prepared as described by Singh et al. [22]. Freshly grown *R. solanacearum* was inoculated into 50 mL BG media broth with 0.5% glucose and incubated at 28 °C and 150 rpm for 24 h. The bacterial cultures were obtained by centrifugation at 4000 rpm and 4 °C for 15 min and were then resuspended in an equal volume of sterile distilled water to obtain a concentration of approximately 10^9^ cfu mL^−1^. Strain NEAU-HV9 was cultured in ISP 2 broth on rotary shaker for 3 days at 28 °C and centrifuged at 10,000 rpm. Subsequently, cell pellets were diluted in 0.85% (*w*/*v*) NaCl solution and adjusted to 10^7^, 10^8^ or 10^9^ cfu mL^−1^. Root inoculation of *R. solanacearum* in tomato seedlings was carried out as described by Singh et al. [22]. About 15 to 20 mL of *R. solancearum* inoculum was taken in a sterile container. Tomato seedlings (6 to 7 days old) were picked one at a time from the germinated seedling tray and then the roots of each seedling were dipped in the bacterial inoculum (up to the root-shoot junction). Four treatments were established as follows: TR 1 (tomato seedlings were pre-inoculated with suspension (10^7^, 10^8^ or 10^9^ cfu mL^−1^) of strain NEAU-HV9 and then inoculated with *R. solanacearum)*; TR 2 (tomato seedlings were pre-inoculated with *R. solanacearum* and then transferred to microfuge tubes with the addition of 1 to 1.5 mL of sterile water and active fraction, where the final treatment concentrations were 1 × MIC and 2 × MIC, respectively); CK 1 (tomato seedlings were inoculated with sterile water); and CK 2 (tomato seedlings only were inoculated with *R. solanacearum*). For all of the treatments, the root-dip inoculated seedlings were transferred to an empty 1.5 mL sterile microfuge tube. After approximately 5 minutes, 1 to 1.5 mL of sterile water was added to the tube. All the inoculated seedlings, along with the controls, were transferred to a growth chamber maintained at 28 °C with 75% relative humidity (RH) and a 12-h photoperiod. Seedlings were analyzed for disease progression after 7 days. Sets of 4 seedlings were recruited in each dilution inoculation, and each assay was performed in triplicate.

### 2.10. Pot Culture Experiments

Prior to use, seed surfaces were disinfected with 2% sodium hypochlorite for 2 min [46]. Both germination and plant growth conditions followed 75–90% RH and a 12-h photoperiod at 28 °C. Four treatments were established as follows: TR 1 (one day before transplanting the test plants, strain NEAU-HV9 was added into the sterilized soil so that each gram of soil received about 1 × 10^9^ cfu g^−1^ bacterial cells. Seven day old tomato seedlings were transferred to the soil; after three weeks, plants were inoculated with a suspension (OD_600_ = 0.3) of *R. solanacearum*); TR 2 (seven day old tomato seedlings were transferred to sterilized soil; after three weeks, tomato seedlings were irrigated with a solution of actinomycin D (0.6 mg L^−1^). After one day, tomato plants were inoculated with a suspension (OD_600_ = 0.3) of *R. solanacearum* by pouring it onto the soil of unwounded plants at a final concentration of 1 × 10^7^ cfu g^−1^ of soil [47]); CK 1 (seven day old tomato seedlings were transferred to sterilized soil; after three weeks, tomato seedlings were irrigated with sterilized water as positive control); and CK 2 (seven day old tomato seedlings were transferred to sterilized soil; after three weeks, tomato seedlings were inoculated with a suspension (OD_600_ = 0.3) of *R. solanacearum* as a negative control). All plants were kept in the greenhouse at 24–28 °C and 75–90% RH with a 12-h photoperiod. Treatments were replicated three times with five plants per replication. The disease incidence was rated using the 0–4 scale [48].

## 3. Results

### 3.1. Isolation and Identification of an Antimicrobial Compound Producing Strain

More than 20 isolates from the soil samples were isolated, purified, and screened for bioactivity against *R. solanacearum*. Among them, only four isolates showed bioactivity against *R. solanacearum*. Since the methanol extract of the the cell pellet and supernatant of one isolate, designated as NEAU-HV9, revealed a higher activity (30.5 mm and 32.8 mm) against the tested bacterial strain (Table 1, Appendix A), this strain was selected for further studies.

Strain NEAU-HV9 was aerobic, Gram-stain positive and formed well-developed, branched substrate hyphae and aerial mycelium that differentiated into spiral spore chains with oval spores (Figure 1). The spore surface was wrinkled. It had good growth on ISP 1, ISP 2, ISP 3, ISP 4, ISP 5, ISP 6, ISP 7, Bennett’s agar and Nutrient agar, and poor growth on Czapek’s agar (Appendix A). The data on the growth characteristics of NEAU-HV9 in different media are given in Appendix A.

Further characterization of NEAU-HV9 was performed by evaluating various biochemical tests (Appendix A). Growth at 15 °C to 37 °C (optimum: 28 °C) and in the range of pH 5 to 9 (optimum: pH 7.0). Tolerate up to 7% (*w*/*v*) NaCl in the culture medium. Positive for hydrolysis of starch, production of H_2_S, hydrolysis of aesculin and decomposition of adenine, hypoxanthine, tyrosine and xanthine, negative for reduction of nitrate, coagulation and peptonization of milk, liquefaction of gelatin and decomposition of casein. D-Glucose, D-maltose, D-mannitol, D-galactose, inositol, D-mannose, L-rhamnose and D-sucrose are utilized as sole carbon sources, but not L-arabinose, dulcitol, D-fructose, lactose, D-ribose, D-sorbitol or D-xylose. L-Alanine, D-arginine, L-asparagine, L-aspartic acid, L-glutamic acid, L-glutamine, glycine, L-proline, L-serine, L-threonine and L-tyrosine are utilized as sole nitrogen sources, but not creatine. The above growth data of isolate NEAU-HV9 denote that the isolate has the typical characteristics of the genus *Streptomyces*.

Recently, it has been suggested that the 16S rRNA gene can be used as a reliable molecular clock due to 16S rRNA sequences from distantly related bacterial lineages having similar functionalities [49]. Basically, the 16S rRNA gene sequence, comprising of about 1500 bp with hyper variable and conserved regions, is universal in all bacteria. According to Woese’s report [50], comparing a stable part of the genetic code could determine phylogenetic relationships of bacteria. The hyper variable regions of the 16S rRNA gene sequences provide species-specific signature sequences, so it is widely used in bacterial identification all over the world. Therefore, the almost-complete 16S rRNA gene sequence (1510 bp) of strain NEAU-HV9 was obtained and has been deposited as MN578143 in the GenBank, EMBL (European Molecular Biology Laboratory) and DDBJ (DNA Data Bank of Japan) databases. BLAST sequence analysis of the 16S rRNA gene sequence indicated that strain NEAU-HV9 was related to members of the genus *Streptomyces*. The EzBioCloud analysis showed that strain NEAU-HV9 was most closely related to *Streptomyces panaciradicis* 1MR-8^T^ and *Streptomyces sasae* JR-39^T^ with a gene sequence similarity of 98.90% and 98.89%, respectively. In conclusion, based on the 16S rRNA gene sequence and the genetic identity of isolate NEAU-HV9, the isolated strain was further identified by neighbor-joining tree (Figure 2), and was also found to belong to the genus *Streptomyces*.

### 3.2. Structural Characterization of Compound

The active component was isolated from fermentation medium (7.5 L) and one bioactive compound was obtained as red, amorphous powder. The compound had UV visible spectra at 215 nm, 440 nm in methanol. The compound showed absorptions at 220 nm and 254 nm with a retention time of 12.367 min (Appendix A), similar to that of actinomycin class of compounds [51,52]. The structure of the compound was further elucidated by ^1^H NMR, ^13^C NMR, and MS analysis as well as comparison with previously reported data. The ESI-MS of the isolated compound revealed molecular ion peaks at m/z 1277.6 [M+Na]^+^ (Appendix A), which was identical to that of actinomycin D [53]; ^1^H and ^13^C spectra of the isolated compound in CD_4_O also showed great similarities to that of actinomycin D [52,53] (Appendix A). In addition, the retention time of commercial actinomycin D (Biotopped, purity: ≥98%) was 12.328 min (Appendix A), and the retention time of compound 2 was 12.367 min (Appendix A). Compound 2 and commercial actinomycin D have similar activity against *R. solanacearum* (Appendix A). The above results showed that the structure of the main active compound was confirmed to be actinomycin D (Figure 3).

### 3.3. Bioactivity of Isolated Compound

#### 3.3.1. Minimum Inhibitory Concentration (MIC)

The minimum inhibitory concentration (MIC) of the antibacterial compound was determined as described by Rathod et al. [45]. The minimum inhibitory concentration of actinomycin D was determined as 0.6 mg L^−1^ (0.48 μmol L^−1^) against *R. solanacearum* (Table 2).

#### 3.3.2. Biological Assays in Tomato Seedlings

The efficacy of the selected antagonist for the control of *R. solanacearum* was evaluated on tomato seedlings (Figure 4 and Appendix A, Table 3). The disease assessment was carried out using the method described in Kumar [23]. For the plants in the TR 1 group, the 10^9^ cfu mL^−1^ suspension of NEAU-HV9 was effective against *R. solanacearum* when compared with the control (CK 2); all seedlings were as healthy as the CK 1 group. The 10^8^ cfu mL^−1^ suspension of NEAU-HV9 showed very weak bioactivity against *R. solanacearum* compared with the control (CK 2); only one seedling was healthy and others were wilted. The 10^7^ cfu mL^−1^ suspension of NEAU-HV9 exhibited no bioactivity against *R. solanacearum*; all seedlings were wilted, the same as the seedlings that were dried.

Actinomycin D was highly effective against *R. solanacearum* in tomato seedlings (Figure 4 and Appendix A, Table 3). It was notable that all seedlings of the control (CK 2) were wilted, however after treatment with actinomycin D at the concentration 1 × MIC and 2 × MIC, none of the seedlings exhibited disease symptoms; the control efficacy of the formulation was 100%.

### 3.4. Pot Culture Experiments

In the pot culture experiments, NEAU-HV9 and actinomycin D effectively suppressed the development of bacterial wilt caused by *R. solanacearum* (Table 4, Appendix A). The negative control treatment had 73.9% relative disease incidence. For strain NEAU-HV9 and actinomycin D, the control efficacies of the formulations were 82% and 100%, respectively.

## 4. Discussion

Soil-borne diseases have caused a significant decline in yield in the monoculture tomato field [3]. *Ralstonia solanacearum* is an important soil-borne bacterial plant pathogen which is distributed all over the world [5]. Recently, the biological control of soil-borne diseases has attracted more attention due to its environmental friendliness and high efficiency [54]. Therefore, isolation, screening and application of highly efficient antagonistic microorganisms is a key factor in biological control. With this outlook, a *Streptomyces* sp. strain NEAU-HV9 was isolated and found to exhibit antibacterial activities against *R. solanacearum* in the present study. By using 16S rRNA gene sequence analysis, combined with morphological, cultural and physiological characteristics, the results showed that strain NEAU-HV9 belongs to members of the genus *Streptomyces* and was most closely related to *Streptomyces panaciradicis* 1MR-8^T^ and *Streptomyces sasae* JR-39^T^ with gene sequence similarities of 98.90% and 98.89%, respectively.

Actinobacteria, particularly *Streptomyces*, are ubiquitous in the rhizosphere soil and can protect plant from pathogenic fungi/bacteria [55], so they have always been used in agriculture [56]. For instance, several *Streptomyces* species such as strains CAI-24, CAI-121, CAI-127, KAI-32 and KAI-90 have been used as BCAs against *Fusarium* wilt in chickpea plants [57]. The *Streptomyces* sp. CB-75, selected from banana rhizosphere soil, showed antifungal activity against 11 plant pathogenic fungi [54]. In this study, the *Streptomyces* sp. NEAU-HV9 exhibited strong antagonistic activity against *R. solanacearum*. According to the study of Singh et al., susceptibility of early stages of tomato seedlings toward the pathogen was confirmed by root-inoculation of *R. solanacearum* in early stages of tomato seedlings [22]. The antagonistic strains should reach a certain amount to demonstrate a significant biocontrol effect [58,59]. In this study, we inoculated very high numbers of *R. solanacearum* and very high levels of *Streptomyces* sp. NEAU-HV9 (10^9^ cfu mL^−1^) in small tubes in the TR 1 group. After culturing for seven days, all tomato seedlings were as healthy as the CK 1 control group (Table 3). There are only *R. solanacearum* and *Streptomyces* sp. NEAU-HV9 in this artificial system, which can better prove that a single NEAU-HV9 was able to be effective against *R. solanacearum*. On the seventh day, more than 90% of seedlings inoculated with *R. solanacearum* were found to be killed, but water-inoculated control seedlings were not wilted/dried [23]. Freshly grown tomato seedlings are too small to carry out detailed disease assessment, and can only be described as healthy, healthy wilted or dried. In the tests of this study, tomato seedlings inoculated with suspension (10^7^ or 10^8^ cfu mL^−1^) of NEAU-HV9 and *R. solanacearum* showed healthy wilted and dried disease phenotypes at different levels, while all tomato seedlings inoculated with suspension (10^9^ cfu mL^−1^) of NEAU-HV9 and *R. solanacearum* were healthy (Figure 4). The results indicated that there are only *R. solanacearum* and *Streptomyces* sp. NEAU-HV9 in this artificial system, which can better prove that a single NEAU-HV9 was able to be effective against *R. solanacearum*. In addition, strain NEAU-HV9 effectively controlled *R. solanacearum* on larger plants in pot culture experiments (Table 4). Thus, the test presented in this study is viable for a preliminary screening of antagonistic actinobacterial strains against *R. solanacearum* and has important aspects with respect to reduced time, space consumption and economics. Meanwhile, the results showed the possibility of using *Streptomyces* sp. NEAU-HV9 as bioinoculant for *R. solanacearum*.

A wide range of bioactive secondary metabolites with anti-inflammatory, antibacterial, antifungal, antialgal, antimalarial and anticancer activities were produced by actinomycetes. Actinomycetes have produced about two-thirds of available antibiotics that have great practical value [60,61]. For example, *Streptomyces* TP-A0595 produced an antagonist that was determined as 6-prenylindole and effective against *Alternaria brassicicola* by inhibiting the formation of infection hyphae [62]. *Streptomyces griseus* H7602 produced a monomer compound that has suppressive effect on infection by *Phytophthora capsici* [63]. Some well-known antibiotics have been isolated from *Streptomyces* and used as fungicides. Many types of antibiotics with high antibacterial activity were produced from *Streptomyces spectabilis*, including streptovaricin [64], desertomycin [65] and spectinomycin [66], and they have high application value in the pharmaceutical industry [67]. Actinomycin D (or Dactinomycin) is a proverbial antitumor-antibiotic drug, which belongs to the actinomycin family and was isolated from *Streptomyces*. Actinomycin D has been demonstrated to have various biological activities. Gram-negative bacteria were largely inhibited by using 10–100 mg per liter concentrations [68]. Actinomycin D produced by the bacterium *Streptomyces hydrogenans* IB310 was effective against both bacterial and fungal phytopathogens [51], and the authors proposed that actinomycin D might be developed as an antibacterial agent used in agriculture. However, there are no reports on antibacterial activities against *R. solanacearum* and it is not currently used in agriculture. In this study, *Streptomyces* sp. NEAU-HV9, which showed strong antibacterial activity against *R. solanacearum*, was isolated and identified. To learn more about the chemical nature of the antibacterial activity of the culture filtrate, the active compound actinomycin D was finally obtained. In this paper, we tested the in vitro antibacterial activity of actinomycin D against *R. solanacearum* and obtained a MIC value of 0.6 mg L^−1^ (0.48 μmol L^−1^), which was many fold lower than other reported new natural antibacterial agents [69], synthesized antibacterial agents and those of commercial fungicides including gentamicin and streptomycin [70]. The antibacterial activity of actinomycin D against *R. solanacearum* tomato seedlings treated with 1 × MIC and 2 × MIC were determined. None of the seedlings inoculated with actinomycin D exhibited disease symptoms and the phytotoxic rating of actinomycin D was similar to that of a water control. Thus, actinomycin D was not phytotoxic at a concentration of 0.6 mg L^−1^ (0.48 μmol L^−1^). The results suggest that actinomycin D might be useful as a candidate pesticide for the treatment of *Ralstonia solanacearum* in tomato.

## 5. Conclusions

In summary, this study found that *Streptomyces* sp. NEAU-HV9 exerted significant antibacterial activity against *R. solanacearum*, and actinomycin D, which was produced by *Streptomyces* sp. NEAU-HV9, exhibited a minimum inhibitory concentration (MIC) against *R. solanacearum* of 0.6 mg L^−1^ (0.48 μmol L^−1^). In addition, *Streptomyces* sp. NEAU-HV9 and actinomycin D can effectively inhibit the occurrence of *R. solanacearum*. From the results, it is obvious that *Streptomyces* sp. NEAU-HV9 is an important microbial resource as a biological control against *R. solanacearum* and actinomycin D is a promising candidate for the development of potential antibacterial biocontrol agents.

## Figures and Tables

**Figure 1 microorganisms-08-00351-f001:**
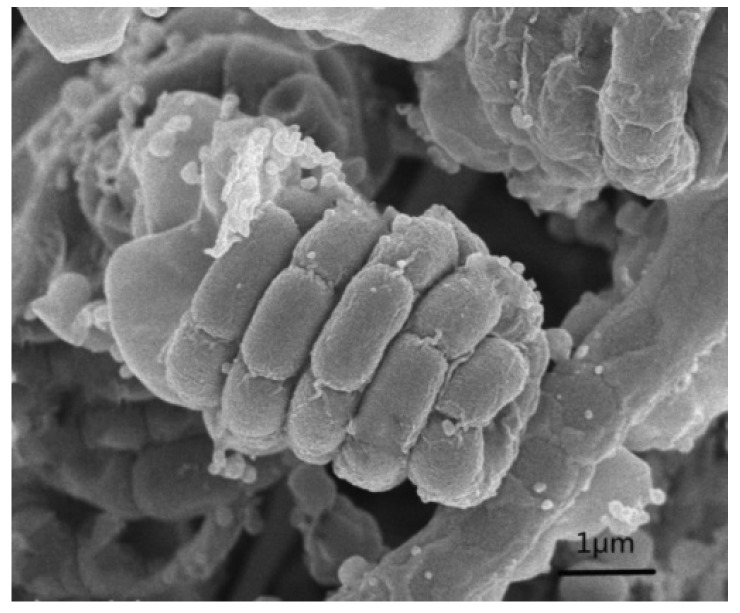
Scanning electron micrograph of strain NEAU-HV9 grown on International *Streptomyces* Project (ISP) 3 agar for 2 weeks at 28 °C.

**Figure 2 microorganisms-08-00351-f002:**
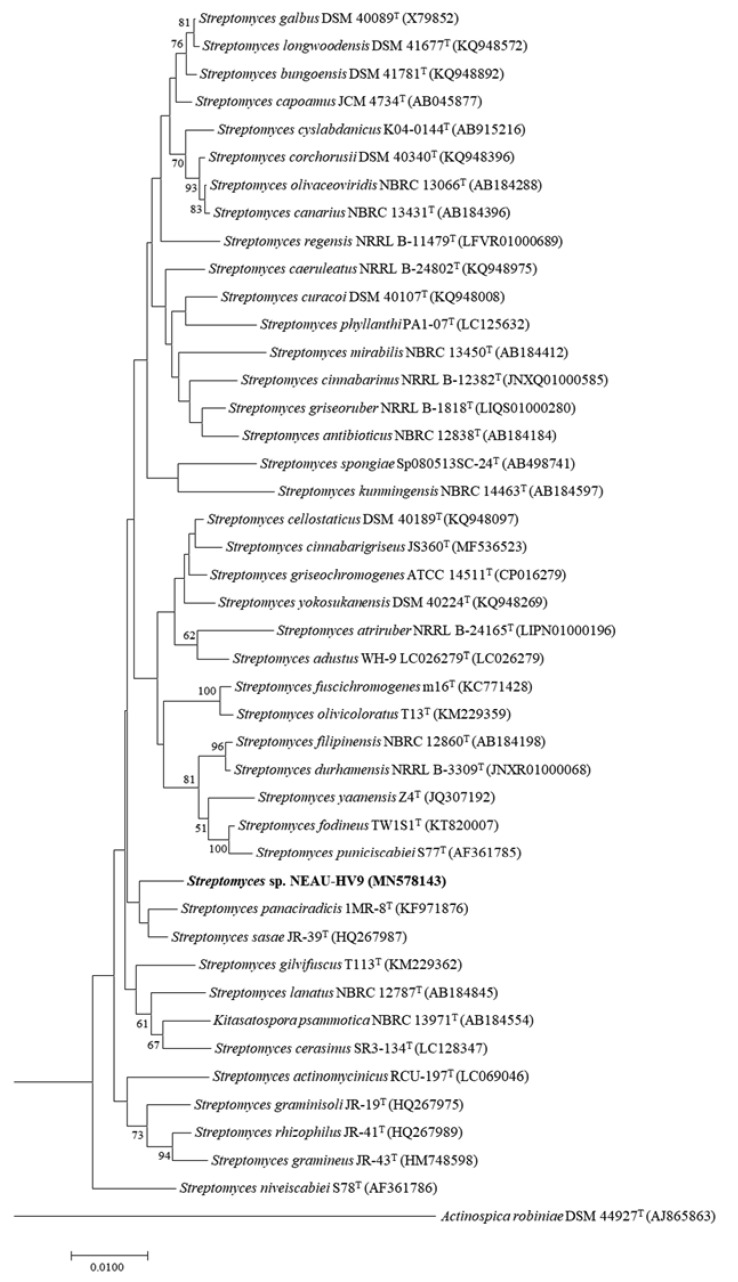
Neighbor-joining phylogenetic tree based on 16S rRNA gene sequences showing the relationships among strain NEAU-HV9 (bold) and members of the genus *Streptomyces*. Bootstrap percentages (≥50%) based on 1000 resamplings are listed at the nodes. *Actinospica robiniae* DSM 44927^T^ was used as the out-group. Scale bar represents 0.01 nucleotide substitutions per site.

**Figure 3 microorganisms-08-00351-f003:**
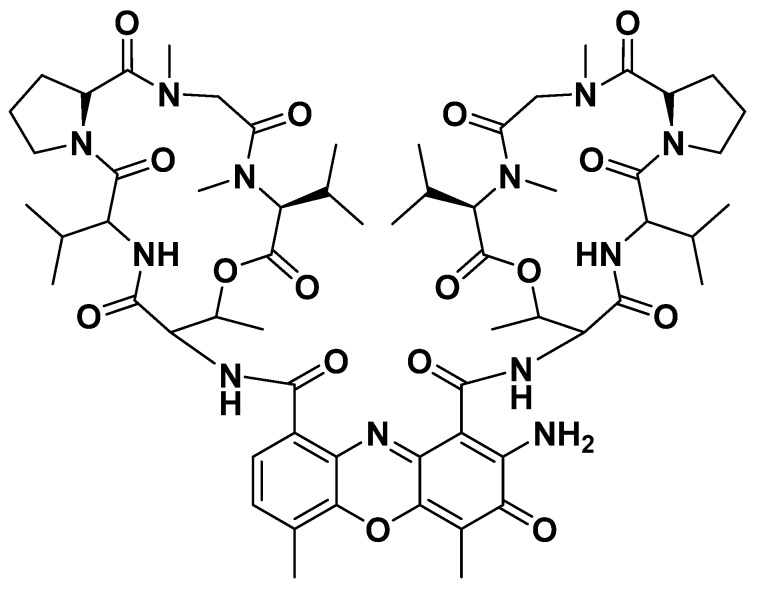
Chemical structure of actinomycin D.

**Figure 4 microorganisms-08-00351-f004:**
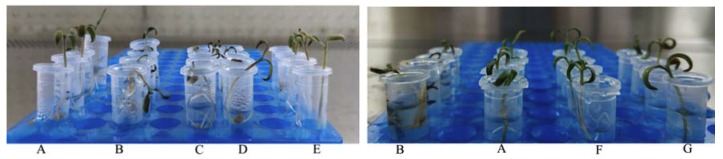
Control efficiency of strain NEAU-HV9 against *R. solanacearum*. A, tomato seedlings were inoculated with sterile water (CK 1); B, tomato seedlings only were inoculated with *R. solanacearum* (CK 2); C, tomato seedlings were pre-inoculated with suspension (10^7^ cfu mL^−1^) of NEAU-HV9 and then inoculated with *R. solanacearum* (TR 1); D, tomato seedlings were pre-inoculated with suspension (10^8^ cfu mL^−1^) of NEAU-HV9 and then inoculated with *R. solanacearum* (TR 1); E, tomato seedlings were pre-inoculated with suspension (10^9^ cfu mL^−1^) of NEAU-HV9 and then inoculated with *R. solanacearum* (TR 1); F, actinomycin D at the concentration 1 × MIC (TR 2); G, actinomycin D at the concentration 2 × MIC (TR 2).

**Table 1 microorganisms-08-00351-t001:** Bioactivities of the supernatant and cell pellet of NEAU-HV9 against *R. solanacearum*.

	Methanol Extract of Cell Pellet	Supernatant
Inhibitory zone diameters (mm)	30.5	32.8

Data shown are the mean of two replications.

**Table 2 microorganisms-08-00351-t002:** Minimum inhibitory concentration (MIC) values of actinomycin D against *R. solanacearum*.

Pathogen	MIC (mg/L)
*R. solanacearum*	0.6 ± 0.2

Data shown are the mean of three replications.

**Table 3 microorganisms-08-00351-t003:** Effect of the strain NEAU-HV9 and actinomycin D on the incidence and control of tomato bacterial wilt in tomato seedlings.

Treatment	Wilt Incidence (%)	Control Efficacy (%)
NEAU-HV9 (10^7^ cfu mL^−1^)	100 ± 0 ^a^	0 ± 0 ^a^
NEAU-HV9 (10^8^ cfu mL^−1^)	93 ± 11.6 ^b^	6.6 ± 11.5 ^b^
NEAU-HV9 (10^9^ cfu mL^−1^)	0 ± 0 ^c^	100 ± 0 ^c^
Actinomycin D (1 × MIC)	0 ± 0 ^c^	100 ± 0 ^c^
Actinomycin D (2 × MIC)	0 ± 0 ^c^	100 ± 0 ^c^
Control	100 ± 0 ^a^	…

Data shown are the mean of three replications. Means within the same column followed by the same letter are not significantly different (*p* = 0.05) according to Fisher’s least significant difference test.

**Table 4 microorganisms-08-00351-t004:** Effect of the strain NEAU-HV9 and actinomycin D on the incidence and control of tomato bacterial wilt in pot culture experiments.

Treatment	Wilt Incidence (%)	Control Efficacy (%)
NEAU-HV9	13.3 ± 5.8 ^b^	82 ± 6 ^a^
Actinomycin D	0 ± 0 ^b^	100 ± 0 ^a^
Control	73.9 ± 6.6 ^a^	

Wilt incidence (WI) was calculated as the percentage of leaves that were completely wilted. Control efficacy was calculated using the following formula: control efficacy (%) = 100 × (WI of control − WI of treatment)/WI of control. Data shown are the mean of three replications. Means within the same column followed by the same letter are not significantly different (*p* = 0.05) according to Fisher’s least significant difference test.

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
