# Peer review of "A Streptomyces sp. NEAU-HV9: Isolation, Identification, and Potential as a Biocontrol Agent against Ralstonia solanacearum of Tomato Plants"

_microorganisms, 2020, doi:10.3390/microorganisms8030351_

Round 1

Reviewer 1 Report

The authors report on the isolation of the well known Streptomyces metabolite actinomycin D (act-D) and its inhibition of the phytopatogen R. Solanacearum.

MIC value given in the abstract and in the text must be indicated also in microMolar, so that to have an easier comparison with other active agents, that must be added in the discussion.

Act-D is described as the first report as inhibitor of plant pathogenic bacteria, but by bibliographic research the following reference has been found:

Zhang, Lili; Wan, Chuanxing; Luo, Xiaoxia; Lv, Lingling; Wang, Xiaopu; Xia, Zhanfeng.Manufacture of actinomycin D with Streptomyces mutabilis for controlling plant pathogenic bacteria, cow mastitis, and female colpitis; Accession Number: 2015:528892, CAN 162:499960

Regarding the isolation of act-D, the crude extract weight from which 53.5 mg of metabolites were isolated must be introduced; this result on the quantitative evaluation of act-D has to be compared with other isolation from Streptomyces reported in literature, discussing the eventual parameters affecting its production.

 A more detailed description of act-D purification must be added; in fact this metabolite is usually produced along with other members of actinomycin class.  The author must describe how pure act-D was evaluated ( e.g. by an analytical HPLC injection, or better by an online LC-ESIMS analysis) and polarimetric value must be  acquired and compared with reported value for the pure compound., Besides signals for [M+H]+ and [M+Na]+ In the MS spectrum in Supplementary, the signal at m/z 1291.6 (not fitting for [M+ K]+ ion) must be assigned.

Reviewer 2 Report

This paper describes the isolation of Streptomyces strains from soil, the characrterisation of in vitro activity against Ralstonia solanacearum, the identification of strain NEAU-HV9 and the identification of the active component, actinomycin D. Streptomyces have previously been reported to produce actinomycin D and previous reports have described actinomycin D activity against bacterial, fungal and viral plant diseases. The authors appear correct in describing this as the first report of actinomycin D inhibiting R. solanacearum on plants, however, the results presented are not conclusive that the strain or compound are promising for disease control, as discussed below.

There is no data presented on the results of the in vitro inhibition assay they used for screening or for the seedling assay using the strain or metabolite. There is no information on how the seedling assay was arranged, i.e. replication, randomisation, disease assessment, statistical analysis etc.. It also appears only one experiment was carried for the strain and metabolite. At a minimum the experiment needs to be repeatable. The photos of results, Figs 5 &6 are interesting but are not data.

The seedling disease assay is a very artificial system in small tubes inoculated with very high numbers of R. solanacearum, as well as with very high levels of the Streptomyces. There is no discussion on how this would relate to infection in commercial plantings, or the economics of using such high levels of biocontrol agent. The test presented is OK for a preliminary screening but a more detailed experiment on larger plants would be required to provide substantive evidence of potential usefulness.

One of the problems with R. solanacearum control is its ability to rapidly become resistant to bacteriocides, biocontrol agents and bacterial phages. How easily does R. solanacearum become resistant to actinomycin D?

Discussion on actinomycin D in relation to control of plant diseases is lacking. The reference to Waksman and Woodruff (66) is 80 years old. A preliminary literature search produces later research, e.g. Kulkarni et al 2017 https://doi.org/10.1016/j.bcab.2017.02.009, describes activity against of actinomycin D against a number of bacterial and fungal pathogens.

Other minor corrections

L5. There is no institute allocation for Wu.

In vitro and in vivo should be in italics

Species name should be in lower case

L72-75. There is no rational as to why this site was sampled. This should be provided

What is BG medium, spell out at first use.

L158 Cells should be lower case

L175 spell out number at start of sentence

L190, change to “More than 20 isolates from the soil samples …”

Fig 4. How was this chemical structure determined, was this determined from this study of copied?

Author Response

Dear Reviewer,

Thank you for the valuable suggestions. In the following, we provide our itemized list of changes according to your suggestions.

Thank you very much for your kindness and help.

Sincerely yours,

Wensheng Xiang

This paper describes the isolation of Streptomyces strains from soil, the characrterisation of in vitro activity against Ralstonia solanacearum, the identification of strain NEAU-HV9 and the identification of the active component, actinomycin D. Streptomyces have previously been reported to produce actinomycin D and previous reports have described actinomycin D activity against bacterial, fungal and viral plant diseases. The authors appear correct in describing this as the first report of actinomycin D inhibiting R. solanacearum on plants, however, the results presented are not conclusive that the strain or compound are promising for disease control, as discussed below.

There is no data presented on the results of the in vitro inhibition assay they used for screening or for the seedling assay using the strain or metabolite. There is no information on how the seedling assay was arranged, i.e. replication, randomisation, disease assessment, statistical analysis etc. It also appears only one experiment was carried for the strain and metabolite. At a minimum the experiment needs to be repeatable. The photos of results, Figs 5 &6 are interesting but are not data.

Reply: The experiment was repeatable, and three experiments were carried. Figs 5 & 6 are only one of them. We didn’t describe it in Materials and Methods, I am so sorry for my mistake and carelessness. We presented detailed description of replication in Materials and Methods. As described by Kumar et al., we presented description on the results of the disease assessment in Result and Discussion parts as follow: (Please see lines 210-211, 281-287 and 335-340). Freshly grown tomato seedlings are too small to carry out detailed disease assessment, and can only be described as healthy, healthy wilted or dried. In the tests of this study, tomato seedlings inoculated with suspension (107, 108 cfu mL-1) of NEAU-HV9 and R. solanacearum showed healthy wilted and dried at different levels, but all of tomato seedlings inoculated with suspension (109 cfu mL-1) of NEAU-HV9 and R. solanacearum were healthy (Figure. 5).

The seedling disease assay is a very artificial system in small tubes inoculated with very high numbers of solanacearum, as well as with very high levels of the Streptomyces. There is no discussion on how this would relate to infection in commercial plantings, or the economics of using such high levels of biocontrol agent. The test presented is OK for a preliminary screening but a more detailed experiment on larger plants would be required to provide substantive evidence of potential usefulness.

Reply: Thank you very much for your valuable suggestion. According to Monteiro and Guidot’s research, tomato seeds are sown to obtain seedlings that take 5-6 days to sprout. Then seedlings are transferred to pots containing soil and grown in greenhouse for about one month. Following this, plants are shifted to growth chamber where plants are inoculated with the pathogen by soil drench or stem inoculation method. In this approach it takes usually 40 days period to perform a single virulence assay. The infection achieved by this way is generally not axenic as soil conditions used are not devoid of other bacterial communities that can colonize the plant during its growth prior to the infection study. The association of other endophytes in turn might pose hindrance in achieving accurate result in sensitive experiments. However, seedling disease assay takes 15 ~ 20 days starting from the seed germination to infection completion. So is less labor intensive for a single infection study. This methodology has important aspects with respect to reduced time, space consumption and economics. Besides, Kumar had found out that the death of tomato seedlings was actually occurring due to R. solanacearum presence in the sterile water. On the basis of previously reported, we took freshly grown tomato seedlings to biological assays. Small tubes only contained R. solanacearuma and NEAU-HV9, result demonstrated that a single NEAU-HV9 was able to be effective against R. solanacearum. In this research, we provided the preliminary evidence that Streptomyces NEAU-HV9 has potential practical value. In the future study, we will take larger plants to a more detailed experiment, which provide substantive evidence of potential usefulness and other related tests also will be performed to evaluate the practicability of NEAU-HV9.

One of the problems with solanacearum control is its ability to rapidly become resistant to bacteriocides, biocontrol agents and bacterial phages. How easily does R. solanacearum become resistant to actinomycin D?

Reply: R. solanacearum causes a vascular wilt disease and has been ranked as the second most important bacterial pathogen. Difficulties are associated with controlling this pathogen due to its abilities to grow endophytically, survive in soil, especially in the deeper layers, travel along water, and its relationship with weeds. Increased incidence of antibiotic-resistant bacteria, lack of development of new antibiotics, and limitations imposed on the use of phytopharmaceuticals in agriculture also aggravates the difficulty of control. The management of bacterial wilt with physical, chemical, biological, and cultural methods has been investigated for decades. Yuliar extensively reviewed R. solanacearum based on findings published between 2005 and 2014. Studying on methods regarding the biological control of bacterial wilt (54%) were the most common, followed by those on cultural practices (21%), chemical methods (8%), and physical methods (6%). Some studies also focused on integrated pest management (IPM) (11%). It's very difficult to control R. solanacearum effectively in only one way, however, IPM play an important role in an integrated plant disease control program, and reduced bacterial wilt disease by 20–100% in the field or under laboratory conditions, and typically combines two or three methods among cultural practices and chemical and biological methods. In this study, we provide the preliminary evidence that strain NEAU-HV9 and act-D were effective against R. solanacearum, which provide the basis for the integrated pest management in the future. Biological control, compared with the chemical methods, has no pollution and it’s not easy to produce resistance, benefit to human and animal safety and environmental protection. We pay more attention to the biological control of bacterial wilt in this study. Thank you very much for your valuable suggestion.

Discussion on actinomycin D in relation to control of plant diseases is lacking. The reference to Waksman and Woodruff (66) is 80 years old. A preliminary literature search produces later research, e.g. Kulkarni et al 2017 https://doi.org/10.1016/j.bcab.2017.02.009, describes activity against of actinomycin D against a number of bacterial and fungal pathogens.

Reply: Thank you very much for your valuable suggestion. We have added the discussion on actinomycin D in relation to control of plant diseases and also the new reference. (Please see lines 360-362 and 366-370)

Other minor corrections

There is no institute allocation for Wu.

We have revised. Thank you. (Please see line 5)

In vitro and in vivo should be in italics

We have revised. Thank you. (Please see lines 15, 24 and 366)

Species name should be in lower case

We have revised. Thank you.

L72-75. There is no rational as to why this site was sampled. This should be provided

We have revised. Thank you. (Please see line 92)

What is BG medium, spell out at first use.

We have revised. Thank you. (Please see lines 112-113)

L158 Cells should be lower case

We have revised. Thank you. (Please see line 181)

L175 spell out number at start of sentence

We have revised. Thank you. (Please see lines 194-195)

L190, change to “More than 20 isolates from the soil samples …”

We have revised. Thank you. (Please see line 214)

Fig 4. How was this chemical structure determined, was this determined from this study of copied?

Yes, the structure of the compound was elucidated by 1H NMR, 13C NMR, and MS analysis as well as comparison with reference. ESI-MS of isolated compound revealed molecular ion peaks at m/z 1,277.6 [M+Na]+, which was identical to that of actinomycin D.

Round 2

Reviewer 1 Report

The structural characterization of act-D is still not sufficient: the similarity reported for chromatographic retention time, the MS signal for the impurity and the lack of polarimetric data cannot confirm the assignment. In fact, the molecular structure of act-D is complex and there is the possibility to have an isomeric metabolite.

Act-D is commecially available and the  biological data,  acquired using a sample containing impurity.  could be evaluated for a pure sample of commercial compound.

In this form the work cannot be accepted.

Author Response

Dear Reviewer,

Thank you for the valuable suggestions. In the following, we provide our itemized list of changes according to your suggestions.

Thank you very much for your kindness and help.

Sincerely yours,

Wensheng Xiang

The structural characterization of act-D is still not sufficient: the similarity reported for chromatographic retention time, the MS signal for the impurity and the lack of polarimetric data cannot confirm the assignment. In fact, the molecular structure of act-D is complex and there is the possibility to have an isomeric metabolite. Act-D is commecially available and the biological data, acquired using a sample containing impurity. could be evaluated for a pure sample of commercial compound.

Reply: We will answer both your questions 1 and 2 together. Under the same conditions, we ferment another 7.5L. Compound 1, compound 2 and compound 3 produced by the strain NEAU-HV9 were purified by HPLC (tR 10.928 min, tR 12.367 min and tR 13.570 min), and were tested for antibacterial activity against R. solanacearum (figure 3 and figure 4). We found compound 1 and compound 2 had similar antibacterial activities and are much better than compound 3. In figure 1, we find that the production of compound 2 is much more than component 1. Based on the above results, we choose compound 2 for subsequent research.

We purchased commercial act-D and performed HPLC analysis. The retention time of commercial act-D was 12.328 min (figure 2), and was analogous to compound 2 produced by the strain NEAU-HV9 (figure 1). Commercial act-D and compound 2 produced by the strain NEAU-HV9 had similar antibacterial activities (figure 5). Besides, MS analysis as well as comparison with previously reported data: ESI-MS of isolated compound revealed molecular ion peaks at m/z 1,277.6 [M+Na]+ (Figure S3).

In summary, we have reason to think that compound 2 is act-D. During the purification process, although we did not completely purify compound 2 which contained very few impurities, by retention time and mass spectrometry, its main metabolite can be determined as act-D. Thank you very much for your valuable suggestion. (Please see lines 162-174, 273-285, Figure S2, Figure S6 and Figure S7)

Figure 1—5,please see the attachment. Fgiure S2, S3, S6 and S7 are in the supplementary file. Thank you very much.

Reviewer 2 Report

Although interesting and the authors have made changes I am still not convinced this is worth publishing in the current form as detailed below.

There is still no data. The authors response was that the experiment was repeated 3 times, but still only one photo for the strain and one for metabolite.

The authors claim the strain or metabolite are have potential for control of Ralstonia, however, this is only a preliminary screen in a very artificial system. If this study was to present a rapid screening method, the authors need to show that this reflects control on a larger plant under conditions reflecting production systems, whether in hydroponics or in soil. If the paper is to present a prospective biocontrol agent they need to show control on mature plant. It is a long way from a barely emerged seedling system to control on a plant.

Author Response

Dear Reviewer,

Thank you for the valuable suggestions. In the following, we provide our itemized list of changes according to your suggestions. All the figures  are in the attachment.

Thank you very much for your kindness and help.

Sincerely yours,

Wensheng Xiang

Although interesting and the authors have made changes I am still not convinced this is worth publishing in the current form as detailed below.

There is still no data. The author response was that the experiment was repeated 3 times, but still only one photo for the strain and one for metabolite.

Reply: We have provided the other two result photo for the strain and metabolite below. We only put only one photo for the strain and one for metabolite in the manuscript. Thank you very much for your valuable suggestion.

Figure: Control efficiency of strain NEAU-HV9 against R. solanacearum. A, tomato seedlings were inoculated with sterile water (CK 1); B, tomato seedlings only were inoculated with R. solanacearum (CK 2); C, tomato seedlings were preinoculated with suspension (107 cfu mL-1) of NEAU-HV9 and then inoculated with R. solanacearum (TR 1); D, tomato seedlings were preinoculated with suspension (108 cfu mL-1) of NEAU-HV9 and then inoculated with R. solanacearum (TR 1); E, tomato seedlings were preinoculated with suspension (109 cfu mL-1) of NEAU-HV9 and then inoculated with R. solanacearum (TR 1).

Figure: Control efficiency of the actinomycin D against R. solanacearum. A, tomato seedlings were inoculated with sterile water (CK 1); B, tomato seedlings only were inoculated with R. solanacearum (CK 2); F, actinomycin D at the concentration 1 × MIC (TR 2); G, actinomycin D at the concentration 2 × MIC (TR 2).

The authors claim the strain or metabolite are have potential for control of Ralstonia, however, this is only a preliminary screen in a very artificial system. If this study was to present a rapid screening method, the authors need to show that this reflects control on a larger plant under conditions reflecting production systems, whether in hydroponics or in soil. If the paper is to present a prospective biocontrol agent they need to show control on mature plant. It is a long way from a barely emerged seedling system to control on a plant.

Reply: Thank you very much for your valuable suggestion. We have done three pot culture experiments on a larger plant in soil in order to prove that NEAU-HV9 is a prospective biocontrol agent. We have provided the other two result photo for the strain and metabolite below. (Please see lines 17-18, 210-222, 322-325, 364-365, 367-369 and Figure 7)

Figure. Control efficiency of the actinomycin D and strain NEAU-HV9 against R. solanacearum. CK 1, positive control; CK 2, negative control; 1, tomato seedlings were preinoculated with suspension (109 cfu g-1) of NEAU-HV9; 2, actinomycin D at the concentration 1 × MIC.

Figure. Control efficiency of the actinomycin D and strain NEAU-HV9 against R. solanacearum. CK 1, positive control; CK 2, negative control; 1, tomato seedlings were preinoculated with suspension (109 cfu g-1) of NEAU-HV9; 2, actinomycin D at the concentration 1 × MIC.

Round 3

Reviewer 1 Report

In their revised version, the authors have taken into account all comments, presenting exhaustive answers and implemented the text.

Author Response

None. Thank you very much!

Reviewer 2 Report

This resubmission still has one major problem, there is no data presented or associated statistical analysis.

Data could be presented on: The initial in vitro screening, the MIC, seedling tube experiment, pot experiment. I again reiterate my previous comments that photos are not data.

It is good that they have included a pot experiment, but the details of this are incomplete. Enough detail should be provided to replicate the experiment, and results of at least 2 independent experiments need to be presented.

Author Response

Thank you very much for the valuable suggestions. In the following, we provide our itemized list of changes according to your suggestions and highlighted the changes in our manuscript.

Thank you very much for your kindness and help.

Sincerely yours,

Wensheng Xiang

This resubmission still has one major problem, there is no data presented or associated statistical analysis.

Data could be presented on: The initial in vitro screening, the MIC, seedling tube experiment, pot experiment. I again reiterate my previous comments that photos are not data.

Reply: We have added data of the initial in vitro screening, the MIC, seedling tube experiment and pot experiment. (Please see Table 1, Table 2, Table 3, Table 4, Figure 4, Figure S2, Figure S9 and Figure S10)

It is good that they have included a pot experiment, but the details of this are incomplete. Enough detail should be provided to replicate the experiment, and results of at least 2 independent experiments need to be presented.

Reply: We have provided more detail about pot experiments to replicate the experiment and the other two results for the strain and metabolite. (Please see lines 210-225, Table 4 and Figure S10)